# Long QT Syndrome Management during and after Pregnancy

**DOI:** 10.3390/medicina58111694

**Published:** 2022-11-21

**Authors:** Agne Marcinkeviciene, Diana Rinkuniene, Aras Puodziukynas

**Affiliations:** 1Department of Cardiology, Lithuanian University of Health Sciences, LT-44307 Kaunas, Lithuania; 2Department of Cardiology, Hospital of Lithuanian University of Health Sciences Kauno Clinics, LT-50161 Kaunas, Lithuania; 3Kaunas Region Society of Cardiology, LT-44307 Kaunas, Lithuania

**Keywords:** long QT syndrome, pregnancy, β-blocker

## Abstract

Long QT syndrome (LQTS) is majorly an autosomal dominantly inherited electrical dysfunction, but there are exceptions (Jervell and Lange-Nielsen syndrome is inherited in an autosomal recessive pattern). This disorder prolongs ventricular repolarization and increases the risk of ventricular arrhythmias, syncope, and even sudden cardiac death. The risk of fatal events is reduced during pregnancy, but dramatically increases during the 9 months after delivery, especially in patients with LQT2. In women with LQTS, treatment with β-blockers at appropriate doses is recommended throughout pregnancy and the high-risk postnatal period. In this review, we summarize the management of LQTS during pregnancy and beyond.

## 1. Introduction

Concomitant cardiac pathology, including arrhythmogenic channelopathies, can complicate the course of a normal pregnancy and could be related to additional risks for both fetus and mother, including sudden cardiac death (SCD). Long QT syndrome (LQTS) is a cardiac channelopathy, characterized by the prolongation of ventricular repolarization and polymorphic ventricular tachycardia (VT) known as “Torsades de pointes”, leading to syncope or SCD. The prevalence of LQTS is approximately 1 in 2000 individuals and can be inherited (approximately 90%) or occur de novo (10%) [1]. LQTS is caused by mutations in several ion-channel genes. The commonest mutations are located in the potassium-channel KCNQ1 (LQT1) and hERG (LQT2) genes, and in the sodium-channel SCN5A (LQT3) gene [2]. These three major genetic subtypes account for approximately 80% of all LQTS cases [3]. There are various subtype-specific triggers for cardiac events in LQTS. Patients with LQT1 experience most of their cardiac incidents as the result of increased adrenergic activity during exercise, especially swimming, or emotional stress. Audible stimulation, such as a ringing telephone or an alarm clock, may provoke LQT2 onset, whereas patients with LQT3 are more likely to have events while resting or sleeping [4]. Women with congenital or acquired LQTS have longer QT intervals and are more likely to develop polymorphic ventricular arrhythmia (VA) or SCD than men [5]. These differences may exist due to alterations of sex hormone levels, dependent on the various menstrual cycle periods, gestation, and the postnatal period, which are related to alterations in QT duration and frequency of cardiac events [6,7].

## 2. Physiological Changes Induced by Pregnancy

Several gradual physiological developments happen during gestation, such as cardiac remodeling and increased cardiac output. Increased heart pulse in pregnancy results in the shortening of QT intervals, therefore it could be protective in patients with LQTS. Conversely the postnatal period is associated with rapid changes in hemodynamics and increased risk of life-threatening arrhythmias. In women with LQTS, the risk of VT and SCD is higher in the postpartum period when compared with the relatively low risk during pregnancy [6]. This may be due to the decrease in the heart rate and the prolongation of the QT interval. It has been discussed that altered sleep patterns, physiological stress, and intense auditory stimuli after childbirth may also contribute to adrenergically mediated cardiac events. The risk of SCD in pregnant women with LQTS was evaluated in a retrospective study [7]. It was reported that the first 40 weeks after giving birth are related to an elevated risk of cardiac disturbances (syncope, aborted cardiac arrest (CA), or SCD) compared with the prepregnancy period of 40 weeks. Moreover, the postpartum period increases the risk for first cardiac events in asymptomatic women with LQTS. Seth et al.’s [6] findings indicate that the 9-month period after childbirth is related to a 2.7-fold elevated risk of a cardiac emergency and a 4.1-fold elevated risk of a life-threatening incidents in comparison to the time before the first conception. After the 9-month postpartum period, the risk of cardiac events reverts to the baseline prepregnancy risk. The risk of cardiac events during pregnancy may vary among different LQTS genotypes. Postpartum cardiac events are more commonly reported in patients with the LQT2 mutation than those with the LQT1 or LQT3 genotypes [6,8].

Changes in sex hormone levels may play a role in modulating cardiac repolarization. Estrogen and progesterone are arrhythmogenic sex hormones and their changes during pregnancy and the postpartum period could potentially provoke cardiac events [9]. Rodriguez et al. [10] reported that in patients with a drug-induced LQTS type, the QT interval length varies in different periods of the menses, with a shorter QT interval during the luteal period when progesterone levels are increased, compared to the follicular period with higher estradiol concentrations. This research showed a proarrhythmic effect of estradiol and an antiarrhythmic effect of progesterone, with a reduced susceptibility to sympathetic stimuli. Unfortunately, we could not find any systematic reviews investigating the impact of different sex hormone levels on QT interval duration in women with congenital LQTS. Odening et al. [11] analyzed the prepubertal ovariectomized transgenic LQT2 rabbits, which were treated with estradiol, progesterone, or placebo. The study showed that progesterone significantly reduced potential triggers for polymorphic VT, such as bigeminy and couplets, and completely eliminated the occurrence of polymorphic VT. In addition, it was observed that progesterone prevents against a SCD, suggesting that high progesterone levels during pregnancy are protective. Moreover, the reduction in progesterone levels during the postpartum period is also associated with postpartum arrhythmias and SCD in patients with LQT2 [8]. Contrariwise, estradiol has an effect by steepening the QT/RR ratio, prolonging cardiac refractoriness, and altering the spatial pattern of dispersion of action potential duration, that results in the promotion of polymorphic VA and SCD. Odening et al. [12] published a case report of a female with LQTS in pregnancy and the postnatal period. Although her medical history suggested LQTS type 2, none of the known mutations were found, except for a polymorphism in KCNE1. During pregnancy and while breastfeeding, her QTc interval length continued to be normalized. The electrocardiogram (ECG) showed prolonged QTc duration when the patient was no longer breastfeeding or on hormone-based contraceptives. The alterations of the ECG indicate a hormonal influence on the QTc duration in women with LQTS. Prospective case-control studies are needed to evaluate the arrhythmogenic effect of sex hormones in pregnant women with LQTS.

## 3. Management of Patients with Long QT Syndrome

Use of β-blockers is the cornerstone of managing congenital LQTS due to its significant reduction in life-threatening arrhythmic risk [13,14]. Their usage significantly reduces a major cardiac event rate during the high-risk postpartum period from 3.7% to 0.8% [6]. According to 2017 American Heart Association (AHA), American College of Cardiology (ACH), and the Heart Rhythm Society (HRS) guidelines on VA and SCD, and the 2018 “European Society of Cardiology (ESC) guidelines” for management of cardiovascular diseases during pregnancy, β-blockers are recommended during and after pregnancy in women with congenital LQTS (recommendation Class I, level of evidence C) [15,16]. Ishibashi et al. [17] in their multicentre study revealed that treatment with β-blockers is essential for the protection of cardiac events during pregnancy and the postnatal period in patients with LQTS. In their study, 136 pregnancies in 76 pregnant women with LQTS were divided into two groups: on β-blocker therapy (β-blockers group) and without β-blocker therapy (non-β-blockers group). During pregnancy and the postnatal period, 14 (11%) cardiac events occurred—all in the non-β-blocker group. Not all β-blockers are the same in the treatment of LQTS. It is shown that the nonselective β-blockers propranolol and nadolol (both are pregnancy risk category C) are significantly more effective than relatively β-1 cardioselective metoprolol (pregnancy risk category C) in preventing cardiac events in symptomatic patients [18]. Furthermore, propranolol has a significantly better QTc shortening effect compared to nadolol and metoprolol, especially in high-risk patients with markedly prolonged QTc. Unfortunately, no randomized clinical trials analyzing the comparative efficacy of different β-blockers for the treatment of LQTS during pregnancy have been conducted [14].

The safety and efficacy of β-blocker use for pregnant women with LQTS and their offspring remains questionable. There is a concern that this treatment may increase the risk of intrauterine grown retardation and malformation in the fetus [17]. In the same study by Ishibashi et al., the frequency of low birthweight infants was significantly higher in the β-blockers group compared with the non-β-blockers group (*p* = 0.024). Although the mean birth weight was within normal range, the birth weight was lower in the β-blockers group than in the non-β-blockers group. This difference between groups may have occurred because the β-blockers group was diagnosed as higher gestational risk, and more than half of the β-blockers group patients had undergone an elective Cesarean delivery before the end of full gestational period. The most used β-blocker in this investigation was propranolol, but events of low birthweight infants did not differ between the different types of β-blockers. Also, there were no significant differences in congenital malformations between the groups. 

Huttunen et al. [19] found that LQT1 patients, exposed to maternal β-blocker treatment during pregnancy, were significantly smaller at birth than patients who weren’t exposed to β-blocker therapy. However, the study showed a rapid catch-up growth during the first year of life. Marshall et al. [20] observed that β-blocker therapy does not increase the risk of fetal distress and miscarriages, even though these infants weigh lower than infants of mothers with LQTS not receiving this therapy. Based on a meta-analysis [21], the first-trimester β-blocker use in general showed no increased odds of any major (non-organ specific) birth defects. On the other hand, examining organ-specific malformations, a 2-fold increase in the risk of cardiovascular defects and an over 3-fold increase in oral clefts and neural tube defects were observed. These data are difficult to interpret due to small numbers of heterogenous studies, lack of statistical significance and potential biases. In addition, β-blockers are secreted in breast milk, however the risk of adverse effects is low in neonates with normal renal and hepatic functions [22]. Hence, it is safe and recommended to continue β-blockers during pregnancy and the postpartum period, at least 40 weeks after delivery [7], in women with LQTS.

Amiodarone (pregnancy risk category D) is contraindicated, as it promotes additional prolongation of the QT interval, herewith it has been related with fetal growth retardation, premature labor, and hypothyroidism [15]. In select LQT3 patients, mexiletine or ranolazine (pregnancy risk category C) can benefit as an add-on therapy to prevent cardiac events in pregnancy. Mexiletine is a class IB antiarrhythmic agent, a late sodium channel blocker that significantly shortens the QT interval [16]. Ranolazine is an antianginal drug used for the treatment of chronic angina. Chorin et al. [23] reported the first long-term study of this drug for LQT3. Ranolazine has been shown to reduce the QT interval of LQT3 harboring the SCN5A-D1790G mutation. There are many genotypic and phenotypic variants in LQT3 that have differing responses to medications, therefore, patients with LQT3 should have a full biophysical assessment of the individual channel mutants and genotypes [24].

The implantation of an implantable cardioverter-defibrillator (ICD) should be considered prior to the pregnancy in women with high-risk factors for a SCD [25]. Conversely, ICD implantation during pregnancy does not increase the risk of major ICD-related problems, such as lead prolapse, lead dysfunction or lead thrombus, under appropriate management and is recommended if an indication emerges [26,27]. Implantation, for ICD ideally single lead, can be done safely, especially if the fetus is beyond 8 weeks of gestation [25]. Programming of the device should be intended to increase the threshold for shock therapies by prolonging the duration of tachycardia episode detection to prevent shock therapies in self-terminating episodes. Immediate electrical cardioversion is recommended for sustained VT regardless of its impact on hemodynamics [16,25].

## 4. Risk Evaluation and Delivery Strategy

For all women with an inherited arrhythmia syndrome, such as LQTS, a multidisciplinary team with expertise in pregnancy, cardiac arrhythmias and genetics should carry out risk assessment and provide pre-pregnancy care [15]. The risk evaluation of LQTS is challenging. The most potent factors predicting CA are LQT2 or LQT3 genotypes, previous events, and degree of QT prolongation [13,16]. QT-interval prolonging drugs and safety data should be checked carefully during pregnancy. Pregnant women suffering from hyperemesis should be closely monitored for electrolyte imbalance, such as hypokalemia and hypomagnesemia. Usage of antiemetic medications may also prolong the QT interval. An ECG, heart ultrasound, and an exercise test are recommended before becoming pregnant. Moreover, Holter monitoring may be useful for a channelopathy. All regular prenatal and peripartum examinations and testing, led by obstetrics, remain an important part of care.

Childbirth may be related to changes in volume, electrolyte disturbances, acute pain, adrenaline release, and the urgent use of antiemetics and anesthetics [28]. The above conditions could complicate the course of childbirth and lead to complications. The 2018 ESC guidelines provide the framework of risk stratification, monitoring and treatment during labor and delivery, which has been developed by expert consensus (see Table 1) [15,28]. The induction of labor should be considered at 40 weeks of gestation. Patients with LQTS, who are at moderate to high-risk, should be supervised at a tertiary center by the expert team, including obstetric and cardiac nurses, an anesthesiologist, a cardiologist with the expertise in inherited arrhythmia syndromes, and an expert obstetrician. In high-risk LQTS, Caesarean delivery is recommended, whereas in low-risk or medium-risk LQTS, the mode of delivery is advised by obstetricians. In the absence of obstetric contraindications, vaginal delivery is considered the safest mode of delivery. According to Tanaka et al. [29], each mode of delivery in women with LQTS has a low likelihood of cardiovascular events, but there is a higher Caesarean delivery rate due to non-reassuring fetal status in labor. It is important to mention that most women (92%) used β-blockers in this study, which may have helped to prevent cardiovascular events during labor. 

Unfortunately, there are no clinical trials that generalize optimal anesthetic management in pregnant women with LQTS, and thus the findings are derived from case reports. In order to avoid life-threatening arrhythmias in LQTS patients, sympathetic activity should be minimized. It is reported that this goal can be achieved with combined spinal-epidural anesthesia [30]. Spinal anesthesia alone is not administered for LQTS patients, as it leads to sudden hemodynamic changes and increased risk of VA and CA [31]. Conversely, epidural anesthesia is associated with gradual alteration in hemodynamics. However, all possible options during childbirth should be discussed and scheduled in advance. For patients in the high-risk LQTS group, continuous ECG monitoring during labor is recommended. It is important to be prepared for intravenous administration of a β-blocker in moderate to high-risk LQTS patients, and for intravenous administration of selected antiarrhythmic medication, such as lidocaine or mexiletine, in case of an emergency in a high-risk LQTS patient (see Table 2). Esmolol, which is rapidly up-titrated and has a short half-life period, can be used if possible. Commonly available alternatives are intermittent intravenous metoprolol and propranolol [28].

## 5. Postpartum Follow Up

So far, there are no general recommendations and approved schemes for how women with LQTS should be supervised after delivery. Different follow up strategies may vary depending on the consensus of the healthcare facility. Women with LQTS should be checked by an experienced cardiologist within the first weeks after delivery and every month for the first 9 months to evaluate efficacy of treatment, ECG, and symptoms [32,33]. Dose adjustment of β -blockers may be needed. Optimal postpartum care remains the same as in routine cases.

## 6. Conclusions

A pregnancy heart team is necessary to evaluate the risk and monitor women with LQTS carefully throughout the pregnancy and delivery. Changes of estradiol and progesterone levels during pregnancy may play a role in arrhythmogenesis. Additional caution may be required in the 9-month postnatal period, which is associated with an increased risk of cardiac events, especially in patients with LQT2. Consequently, the close cardiac follow up with monitoring of ECG during the postnatal period is essential. Treatment with β-blockers is important to protect women with LQTS from cardiac events during pregnancy and the postnatal period. The benefits of β-blockers are outweighed by the risks to the foetus. Genetic counselling and testing for mutations in LQTS patients provide important prognostic and therapeutic information that can help family members understand the risks. However, a negative genetic result does not rule out the diagnosis of LQTS, as has been confirmed by clinical evaluation. In conclusion, pregnancy planning is mostly safe and not contraindicated in LQTS.

## Figures and Tables

**Table 1 medicina-58-01694-t001:** Recommended level of care during labor for women with LQTS. Adapted from the 2018 ESC guidelines [15] and Roston et al. [28].

Risk of Arrhythmia with Hemodynamic Impairment during Labor	LQTS Phenotype	Level of Surveillance	Class ^a^	Level ^b^
Low-risk	LQTS with no previous events and QTc ≤ 470 ms	**1**	I	C
Medium-risk	LQTS with remote events, LQTS with no previous events and QTc ≥ 470 ms	**2**	I	C
High-risk	Any other LQTS with latest incidents *	**3**	I	C
**Descriptions of the planned actions**		**Level of surveillance**		
		**1**	**2**	**3**
Consultation with a cardiologist		+		
Prescribe/terminate the required medication if necessary		+	+	+
Consultation with an expert team at tertiary center			+	+
Method and location of delivery recommended by midwives		+	+	
Labor at thoracic operating theatre (Cesarean delivery advised)				+
Heart rhythm assessment (telemetry, external heart rate monitor)			(+)	+
Intravenous line, administration of β-blocker if necessary			+	+
Arterial line				+
Be prepared to administer selected antiarrhythmic drugs IV				+
Exterior cardioverter-defibrillator on unit			+	+

* Latest incidents are described as arrhythmogenic syncope and/or seizures, CA and/or persistent VA in the last 1 year with adequate medication therapy. CA—cardiac arrest; LQTS—long QT syndrome; ms—milliseconds; QTc—corrected QT interval; IV—intravenous; VA—ventricular arrhythmia. ^a^ Class of recommendation, ^b^ level of evidence.

**Table 2 medicina-58-01694-t002:** Treatment of unstable cardiac rhythm disturbances of LQTS in pregnancy. Adapted from Roston et al. [28].

Therapy for Acute Presentation	Transition to Chronic Therapy
1st line: IV or oral β-blocker	Increase β-blocker ± add mexiletine
2nd line: IV MgSO_4_, lidocaine, mexiletine	Consider K+ supplement ± LCSD (best delayed until post-partum)
3nd line: transvenous pacing *	Consider for ICD indication **

* Before considering a permanent device, a temporary transvenous pacemaker should be attempted. ** The ICD should only be implanted according to strong recommendations. Echocardiography or other non-radiological technique should be used. IV—intravenous; LCSD—left cardiac sympathetic denervation; ICD—implantable cardioverter-defibrillator.

## Data Availability

Not applicable.

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
