# Peer review of "Long QT Syndrome Management during and after Pregnancy"

_medicina, 2022, doi:10.3390/medicina58111694_

Round 1

Reviewer 1 Report

A well-presented and structured review about Long QT syndrome and pregnancy. This subject has been extensively discussed in the past. The paper seems attractive and can be easily comprehended by clinicians and obstetricians who may be interested but not specialised in cardiology.

However, the following minor issues should be considered:  

LQTS is majorly a dominantly inherited condition, but there are exceptions (Jervell and Lange-Nielsen syndrome is a recessive variant). Please amend line 8 as needed.

Several minor issues, especially related to the use of English language, style, and repetitions have also been spotted. A thorough text review is suggested for language refinement. The following changes could be made:

line 20: add 'both' before the phase 'fetus and mother'. Delete 'the risk of' since it is repeated earlier in the sentence.

line 25: omit 'the' in phrase 'caused by the mutations'

line 28: 'various' instead of 'relatively'

line 33: the sentence is unnecessarily repeated in the previous one

line 38: repeated in line 35. please amend the sentence

line 41: 'Several' instead of 'There are many'. 'occur' instead of 'occuring'

line 44, 80: 'Otherwise' should be substituted with a different word, such as 'contrariwise' or 'conversely' 

line 45: 'arrhythmias' instead of 'arrhythmogenic'

line 46-47: repeated in previous sentence. please amend.

line 94: 'in patients with LQTS' could be omitted

line 114: change the phrase 'are done'. 'have been conducted' could be used

line 117: 'In the same study' but no previous reference is mentioned

line 121: 'difference' instead of 'different' 

line 122: higher 'gestational' risk?

line 140: the word 'eventually' could be changed. 'In conclusion', 'Therefore', 'Hence' could be used

line 165: 'criteria' should be changed. The phrase could amended as: 'the above conditions could...'

line 173: 'whereas' instead of 'otherwise'

line 214: ',which' instead of 'and diagnosis'

The References should be rechecked for typos (as 'Heart Rhythm' instead of 'Hear Rhythm', line 222: 'long-QT' instead of 'long-qt', line 275: 28761, line 277: Circulation Journal. Circ J, etc.) and should be uniformly presented according to the journal guidelines.

Author Response

We are very grateful for your comments and clarifications. We have made corrections to the article. Please see the revised manuscript.

Reviewer 2 Report

The authors review an important topic of long QT syndrome during  and after pregnancy.

Although the topic is not novel, it is a well written concise review which has not been addressed before in the journal. However, the review can be improved further and address significant points which have not been touched upon. 

I suggest the following:

1.     These two sentences in introduction is repetition – Please correct

Auditory stimuli such a ringing telephone or an alarm clock may pro- 31 voke LQT2 onset, whereas patients with LQT3 are more likely to have events during rest 32 or sleep [4]. 

Cardiac events triggered by auditory stimuli, such as an alarm clock, tend to 33 occur in LQT2 patients

2.     Anesthetic implications of long QT syndrome during pregnancy should also be briefly discussed in the delivery strategy section

3.     The implantation of an implantable cardioverter-defibrillator (ICD) should be considered prior to the pregnancy in women with high-risk factors for a SCD – please highlight the high risk indications for ICS such as QTc >500 milliseconds, LQT2 and LQT3 (class IIa indication, level of evidence B)

4.     Further, can improve the discussion by highlighting the implication of ICD implantation during pregnancy and shock therapies.

5.     Further discussion about antiarrhythmics other than betablockers should be made to improve the manuscript. 

a.     For example, should highlight that commonly used Amiodarone should not be used during pregnancy (category D), as it has been associated with fetal growth retardation, hypothyroidism, and premature labor. 

b.     Other meds such as Ranolazine and mexiletine and potential use in select LQT3 patients

6.     Discussion of follow up on these patients post-delivery as well as mode of testing should be highlighted especially in the high risk 9 month post-natal period as the authors suggest especially as the title includes after pregnancy, 

Author Response

Point 1:  These two sentences in introduction is repetition – Please correct

Auditory stimuli such a ringing telephone or an alarm clock may pro- 31 voke LQT2 onset, whereas patients with LQT3 are more likely to have events during rest 32 or sleep [4].  Cardiac events triggered by auditory stimuli, such as an alarm clock, tend to 33 occur in LQT2 patients.

Response 1: Auditory stimuli such a ringing telephone or an alarm clock may provoke LQT2 onset, whereas patients with LQT3 are more likely to have events during rest or sleep [4]. The following sentence is deleted.

Point 2: Anesthetic implications of long QT syndrome during pregnancy should also be briefly discussed in the delivery strategy section

Response 2: There are no clinical trials that generalize optimal anesthetic management in pregnant women with LQTS and findings are derived from case reports. In order to avoid life-threatening arrhythmias in LQTS patients, sympathetic activity should be minimized. It is reported that this goal can be achieved with combined spinal-epidural anesthesia [1]. Spinal anesthesia alone is not administered for LQTS patients, as it leads to sudden hemodynamic changes and increaded risk of ventricular arrhythmias and cardiac arrest [2]. Conversely, epidural anesthesia is associated with gradual alteration in hemodynamics.

  1. Hashimoto, E; Kojima, A; Kitagawa, H; Matsuura, H. Anesthetic Management of a Patient With Type 1 Long QT Syndrome Using Combined Epidural-Spinal Anesthesia for Caesarean Section: Perioperative Approach Based on Ion Channel Function. J Cardiothorac Vasc Anesth. 2020 Feb;34(2):465-469
  2. Al-Refai, A; Gunka, V; Douglas, J. Spinal anesthesia for Cesarean section in a parturient with long QT syndrome. Can J Anaesth. 2004 Dec;51(10):993-6.

Point 3:     The implantation of an implantable cardioverter-defibrillator (ICD) should be considered prior to the pregnancy in women with high-risk factors for a SCD – please highlight the high risk indications for ICS such as QTc >500 milliseconds, LQT2 and LQT3 (class IIa indication, level of evidence B)

Point 4:     Further, can improve the discussion by highlighting the implication of ICD implantation during pregnancy and shock therapies.

Response 3, 4: Implantation of ICD during pregnancy carries additional risks related with interventional procedure and possible radiation exposure to the featus. In a case of indications to implant an ICD in patients of high-risk LQTS group, the procedure should be performed before the planned pregnancy. Programming of the device should be intended to increase the threshold for shock therapies by prolonging the duration of tachycardia episode detection to prevent shock therapies in self terminating episodes.

Point 5:     Further discussion about antiarrhythmics other than betablockers should be made to improve the manuscript. 

  1. For example, should highlight that commonly used Amiodarone should not be used during pregnancy (category D), as it has been associated with fetal growth retardation, hypothyroidism, and premature labor. 

Response 5a: Nonselective β-blockers such as atenolol (pregnancy risk category D) should be avoided in pregnant women with LQTS. Welzel and collegues [1] performed a review and reported that atenolol was related with the highest risk for intrauterine growth retardation (IUGR) of all analyzed β-blockers. β-1 cardioselective blockers seem to have less effect on uterine contraction and peripheral vasodilation that lead to lower rate of IUGR [2].

Other antiarrhythmic drugs are not recommended for treatment in patients with LQTS, except for lidocaine (pregnancy risk category C) or mexiletine (pregnancy risk category C), which could safely be used in treatment of unstable cardiac rhythm disturbances. Amiodarone (pregnancy risk category D) is contraindicated, as it promotes additional prolongation of QT interval, herewith it has been related with fetal growth retardation, premature labor, and hypothyroidism [2].

  1. Welzel, T; Donner, B; van den Anker, JN. Intrauterine Growth Retardation in Pregnant Women with Long QT Syndrome Treated with Beta-Receptor Blockers. Neonatology. 2021;118(4):406-415.
  2. Regitz-Zagrosek, V; Roos-Hesselink, JW; Bauersachs, J; Blomström-Lundqvist, C; Cífková, R; De Bonis, M; et al. 2018 ESC Guidelines for the management of cardiovascular diseases during pregnancy. Vol. 39, European Heart Journal. Oxford University Press; 2018. p. 3165–241

Other meds such as Ranolazine and mexiletine and potential use in select LQT3 patients

Response 5b: In select LQT3 patients mexiletine or ranolazine (pregnancy risk category C ) can benefit as add on therapy to prevent cardiac events in pregnancy. Mexiletine is a class IB antiarrhythmic agent, a late sodium channel blocker that significantly shortens QT interval [3]. Ranolazine is an anti-anginal drug used for the treatment of chronic angina. Chorin and others [4] reported the first long-term study of this drug for LQT3. Ranolazine has been shown to reduce QT interval of LQT3 harboring the SCN5A-D1790G mutation. There are many genotypic and phenotypic variants in LQT3 that response to medications differ, therefore patients with LQT3 should have a full biophysical asessment of the individual channel mutants and genotypes [5].

  1. Al-Khatib, SM; Stevenson, WG; Ackerman, MJ; Bryant, WJ; Callans, DJ; Curtis, AB; et al. 2017 AHA/ACC/HRS Guideline for Management of Patients With Ventricular Arrhythmias and the Prevention of Sudden Cardiac Death. Circulation. 2018 Sep 25;138(13):e272–391.
  2. Chorin, E; Hu, D; Antzelevitch, C; Hochstadt, A; Belardinelli, L; Zeltser D; et al. Ranolazine for Congenital Long-QT Syndrome Type III: Experimental and Long-Term Clinical Data. Circ Arrhythm Electrophysiol. 2016 Oct;9(10):e004370.
  3. Lee, MJ; Monteil, DC; Spooner, MT. Peripartum management of patient with long QT3 after successful implantable cardioverter defibrillator device discharge resulting in device failure: a case report. Eur Heart J Case Rep. 2021 Nov 30;5(12):ytab487.

Point 6:     Discussion of follow up on these patients post-delivery as well as mode of testing should be highlighted especially in the high risk 9 month post-natal period as the authors suggest especially as the title includes after pregnancy.

Response 6: Postpartum follow up

So far, there are no general recommendations and approved schemes for how women with LQTS should be supervised after delivery. Diferrent follow up strategies may vary depending on the consensus of the healthcare facility. Women with LQTS should be checked by an experienced cardiologist within the first weeks after delivery and every month for the first 9 months to evaluate efficacy of treatment, ECG and symptoms [1, 2]. Dose adjustment of β -blocker may be needed. Optimal postpartum care remains the same as in routine cases.

  1. Taylor, C; Stambler, BS. Management of Long QT Syndrome in Women Before, During, and After Pregnancy. US Cardiology Review. 2021; 15:e08.
  2. Meregalli, PG; Westendorp, IC; Tan, HL; et al. Pregnancy and the risk of torsades de pointes in congenital long-QT syndrome. Neth Heart J. 2008; 16:422-5.

Reviewer 3 Report

Thank you so much for inviting me to review this interesting article. The authors discussed long QT syndrome management during and after pregnancy. They also pointed out the risk evaluation and delivery strategy for patients with long QT syndrome. I do have the below comments that will improve this review article.

-The authors should comment on the management of QT syndrome during the prenatal period.

-The authors commented on the beta-blocker therapy during pregnancy, but they did not comment on the safety of each beta-blocker therapy during lactation in detail. They only mentioned one line about that area (lines 138-139).

-The authors nicely discussed the management of long QT syndrome, however, they may need to educate the readers about counseling the pregnant woman. E.g. regarding the activity of daily living? any restrictions? Any dietary recommendations? Family support?...etc.

Author Response

We are very grateful for your comments and clarifications. We have made corrections to the article and are sending it for your review. Please see the attachment.

Point 1:  The authors should comment on the management of QT syndrome during the prenatal period.

Response 1: we performed a literature review of pregnant patients with long QT and their care during and after delivery.

Point 2: The authors commented on the beta-blocker therapy during pregnancy, but they did not comment on the safety of each beta-blocker therapy during lactation in detail. They only mentioned one line about that area (lines 138-139).

Response 2: You are completely right, we have not looked at the safety of specific beta-blockers for breastfeeding, apparently because there are numerous articles and webpages dedicated to the safety of drugs during lactation for example: www.infantrisk.com or www.drugs.com.

Point 3:     The authors nicely discussed the management of long QT syndrome, however, they may need to educate the readers about counseling the pregnant woman. E.g. regarding the activity of daily living? any restrictions? Any dietary recommendations? Family support?...etc.

Response 3: This could be an idea for the following article. We agree that usually doctors communicate with the patient regarding assessment of the condition of mother and fetus, prescribing/discontinuing medication, while lifestyle changes are frequently ignored.

Reviewer 4 Report

Overall well written manuscript and important area towards the women health. Please cross check some spelling and references looks miss organised.

Author Response

We are very grateful for your comments and clarifications. We have made corrections in the article and references.

Round 2

Reviewer 2 Report

Authors have improved the manuscript after revision and I have no further comments. 

Reviewer 3 Report

The authors addressed all reviewers' comments.